# Economic evaluation of robot-assisted training versus an enhanced upper limb therapy programme or usual care for patients with moderate or severe upper limb functional limitation due to stroke: results from the RATULS randomised controlled trial

Cristina Fernandez-Garcia  ,[1] Laura Ternent,[1] Tara Marie Homer,[1] Helen Rodgers,[2,3] Helen Bosomworth,[2] Lisa Shaw,[2] Lydia Aird,[3] Sreeman Andole,[4] David Cohen,[5] Jesse Dawson,[6] Tracy Finch,[7] Gary Ford,[2,8] Richard Francis,[1] Steven Hogg,[9] Niall Hughes,[10] H I Krebs,[11] Christopher Price,[2,3] Duncan Turner,[12] Frederike Van Wijck,[13] Scott Wilkes,[14] Nina Wilson  ,[1] Luke Vale[1]

For numbered affiliations see end of article.

**Correspondence to**
Dr Laura Ternent;
laura.ternent@newcastle.ac.uk

## ABSTRACT

**Objective** To determine whether robot-assisted training is cost-effective compared with an enhanced upper limb therapy (EULT) programme or usual care.

**Design** Economic evaluation within a randomised controlled trial.

**Setting** Four National Health Service (NHS) centres in the UK: Queen's Hospital, Barking, Havering and Redbridge University Hospitals NHS Trust; Northwick Park Hospital, London Northwest Healthcare NHS Trust; Queen Elizabeth University Hospital, NHS Greater Glasgow and Clyde; and North Tyneside General Hospital, Northumbria Healthcare NHS Foundation Trust.

**Participants** 770 participants aged 18 years or older with moderate or severe upper limb functional limitation from first-ever stroke.

**Interventions** Participants randomised to one of three programmes provided over a 12-week period: robot-assisted training plus usual care; the EULT programme plus usual care or usual care.

**Main economic outcome measures** Mean healthcare resource use; costs to the NHS and personal social services in 2018 pounds; utility scores based on EQ-5D-5L responses and quality-adjusted life years (QALYs). Cost-effectiveness reported as incremental cost per QALY and cost-effectiveness acceptability curves.

**Results** At 6 months, on average usual care was the least costly option (£3785) followed by EULT (£4451) with robot-assisted training being the most costly (£5387). The mean difference in total costs between the usual care and robot-assisted training groups (£1601) was statistically significant (p<0.001). Mean QALYs were highest for the EULT group (0.23) but no evidence of a difference (p=0.995) was observed between the robot-assisted training (0.21) and usual care groups

(0.21). The incremental cost per QALY at 6 months for participants randomised to EULT compared with usual care was £74 100. Cost-effectiveness acceptability curves showed that robot-assisted training was unlikely to be cost-effective and that EULT had a 19% chance of being cost-effective at the £20 000 willingness to pay (WTP) threshold. Usual care was most likely to be cost-effective at all the WTP values considered in the analysis.

**Conclusions** The cost-effectiveness analysis suggested that neither robot-assisted training nor EULT, as delivered in this trial, were likely to be cost-effective at any of the cost per QALY thresholds considered.

## Strengths and limitations of this study

► Our economic evaluation was designed and conducted following best practice methods which resulted in robust and generalisable results.

► Sensitivity analyses exploring any uncertainties surrounding the level of resource use and their impact on the cost-effectiveness of the interventions add to the robustness of the results.

► The unavailability of longer-term data for the within-trial evaluation means that no robust inferences could be made on the long-term cost-effectiveness of the interventions.

► Poor completion of arm rehabilitation therapy logs meant that detailed information on the delivery of usual care was obtained from the health service utilisation questionnaire.

► The use of quality-adjusted life years based on responses to the EQ-5D-5L questionnaire as a generic outcome measure may not accurately capture changes in quality of life for this patient group.

**Trial registration number** ISRCTN69371850.

## INTRODUCTION

Stroke is the fourth leading cause of death in the UK and a leading cause of disability. Almost two-thirds of patients who had a stroke leave hospital with a disability.[1] A common disability following a stroke is loss of upper limb function. This results in a reduction of the individual's autonomy and impedes activities of daily living. Approximately 80% of people with acute stroke have upper limb motor impairment, with 50% of patients still experiencing problems after 4 years following the stroke.[2 3]

The Robot-Assisted Training for the Upper Limb after Stroke (RATULS) trial sought to evaluate the clinical effectiveness and cost-effectiveness of robot-assisted training (using the MIT-Manus robotic gym system) by comparing it with either an enhanced upper limb therapy (EULT) programme, or usual care.[4 5] Participants in the robot-assisted training and EULT groups also received usual care. The robot-assisted training and EULT programmes were of the same duration and frequency (45 min face-to-face therapy, three times a week for 12 weeks). Results from the trial provided no evidence that robot-assisted training as delivered in the study nor EULT improved upper limb function after stroke compared with usual care.[5]

After conducting a scoping review, we found little evidence of cost-effectiveness studies in the UK. The only economic evaluation we found in the literature assessed the cost-effectiveness of robot-assisted training therapy for upper limb rehabilitation within the USA based VA Robotics study.[6 7] This randomised controlled trial (RCT) also assessed the cost-effectiveness of the MIT-Manus robotic gym system in upper limb rehabilitation in patients who had a stroke. Given the resource intensive nature of stroke rehabilitation programmes[8] and the lifelong impacts of stroke, evidence on the cost-effectiveness of these programmes derived from well-designed economic evaluations is needed. The RATULS trial is, to our knowledge, the largest and first multicentre trial with sufficient statistical power to compare robot-assisted training with another evidence-based therapy programme, or usual care.[9] This paper reports the results from a within-trial analysis that formed part of the RATULS trial.

## TRIAL OVERVIEW
### Summary of RATULS trial

The study was a three-arm RCT which recruited 770 participants from stroke units, day hospitals, outpatient clinics, primary care, community rehabilitation services and local stroke clubs in four study centres. The sample size calculation yielded a target sample size of 762 participants with 216 participants in each group required to provide 80% power at a significance level of 1.7%. The sample size was revised after protocol publication to 770 to allow for 15% attrition (rather than 10% as originally specified in the published protocol). Full details on sample size calculation and trial methodology have been reported elsewhere.[4 5]

Participants were eligible for inclusion in the study if they were 18 years or older, had experienced a first-ever stroke between 1 week and 5 years prior to randomisation and, as a consequence of the stroke, had moderate or severe upper limb functional limitation as measured by the Action Research Arm Test (ARAT)[10] (score 0–39). Potential participants who had been previously enrolled in this study; had participated or were participating in another upper limb rehabilitation study; had previously used the MIT-Manus robotic gym or another arm rehabilitation robot; had other notable upper limb impairment; or had a diagnosis that would interfere with the rehabilitation or outcome assessments were excluded from the trial. All those who were eligible and who consented to participate in the study were randomised on a 1:1:1 ratio to receive one of the three interventions.

The clinical primary outcome of the study was upper limb function 'success' using (ARAT)[10] at 3 months postrandomisation. The definition of success differed depending on baseline severity of upper limb functional limitation. Success for a baseline ARAT score 0–7 was defined as an improvement of 3 points or more; a baseline ARAT 8–13 required an improvement of 4 points or more; baseline ARAT 14–19 required an improvement of 5 points or more and finally, a baseline ARAT 20–39 required an improvement of 6 points or more.[4 5] Full details of the clinical trial results and methodology have been described elsewhere.[4 5]

## Economic evaluation methods

We conducted an economic evaluation consisting of a cost–utility analysis using the quality-adjusted life year (QALY) as the primary outcome measure following guidance for best practice in health technology appraisal.[11]

### Perspective

We conducted all analyses adopting the perspective of the UK National Health Service (NHS) and personal social services setting.

### Costs

The costs included in the analysis comprised the intervention costs for the robot-assisted training and EULT programmes, no intervention costs were directly associated with usual care and we assumed that any rehabilitation received had been reported in the health service utilisation questionnaire. We also included use of health and social services over the 6-month follow-up period for all randomised groups. For the intervention costs we assumed, as per the protocol,[4] that each therapy session in both intervention groups lasted 60 min including 45 min of face-to-face therapy on a 1:1 basis including the same staff component. In addition to staff time, capital costs were calculated for robot-assisted training using the 'equivalent annual cost'[12] methodology applying a 3.5%

discount rate and on the basis that the equipment would need replacing after 5 years.

We developed a health service utilisation questionnaire, informed by previous data collection tools including the Client Service Receipt Inventory[13] to capture health and social care resource use at baseline and 6 months postrandomisation. At baseline, participants were asked about the care received in the 3 months before they joined the study. These data were used to control for any imbalance in participant use of services at baseline. The information collected at 6 months included: visits to accident and emergency departments; outpatient appointments; day patient appointments; overnight hospital stays; the use of general practitioner and nursing services; use of therapy services; medications; community-based healthcare; social services; and residential care and nursing home stays.

To estimate the cost at 6 months for each participant we combined data on use of care and services with unit costs obtained from routine data sources. We applied NHS national reference costs[14] to hospital-based services. Information for staff costs, unit costs relating to primary care, social care and community health services was mostly derived from Curtis and Burns.[15] In order to cost prescribed medication, we collected information on medication name, dosage, frequency and intake duration. We combined this information with unit costs taken from the British National Formulary.[16] When information on intake duration, medication format (eg, dose; mode of intake) was missing, we used prescription cost analysis data[17] and assumed that the participant had been taking the most commonly prescribed format and had been issued one prescription for the trial period.

We used the information collected at 6-month follow-up in addition to the intervention costs to derive the total mean cost per participant per each randomised group. All costs are reported in pounds Sterling and converted to 2018 prices, when appropriate, using the Bank of England inflation calculator.[18]

### Effectiveness outcome

The quality of life outcome measures used for the economic evaluation were the summary utility scores derived from the EQ-5D-5L questionnaires.[19] These were completed by participants at baseline, 3 and 6 months postrandomisation. The EQ-5D-5L questionnaire measures the individual's self-reported health-related quality of life through the following five domains: mobility, self-care, usual activities, pain/discomfort and anxiety/depression. Respondents were asked to describe each domain according to five different levels: no problems, slight, moderate, severe or extreme problems. Since there is no currently accepted valuation set for the EQ-5D-5L questionnaire, we mapped the questionnaire responses to the E5-5D-3L[20 21] descriptive system in order to generate the utility values. These utility values formed the basis of our QALY calculations using the area-under-the-curve approach.[22]

### Missing data

For the main (base-case) cost-effectiveness analysis we only included those participants for whom we had some data on costs. Once we determined that missing cost data were missing at random, we explored in the sensitivity analyses the effect on cost-effectiveness of applying multiple imputation to missing total costs controlling for age, sex and baseline ARAT score.

We explored the patterns of missing utility data. Once we established that information was missing at random, we used multiple imputation methods to estimate the missing utility values at 3 months and 6 months. This involved applying truncated normal regression while controlling for age, sex and baseline ARAT score in order to generate the missing utility value.

### Base-case cost-effectiveness analysis

We calculated mean costs and effects along with corresponding SD. Where we report differences in mean costs and effects between all three randomised groups we used 98.33% CIs, as this was a three-arm comparison. We conducted all pairwise comparisons using 95% CIs. Using seemingly unrelated regression modelling methodology[23] in the adjusted cost-effectiveness analysis, we derived the incremental cost per gained QALY for each participant at 6 months. This approach involved estimating two linear regressions with their own dependent variable for costs and QALYs and a set of explanatory variables. We used randomised group, study centre and time since stroke as explanatory variables for both costs and QALYs. Additionally, we incorporated baseline utility scores as an explanatory variable for the QALY equation and total baseline costs as an explanatory variable to the costs equation. We presented the results from the adjusted analysis in the form of an incremental cost–effectiveness ratio (ICER). The ICER is the difference in mean costs divided by the difference in mean effects (in this case QALYs) between two alternatives.

In the analysis if a comparator was both more costly and less effective than the others it was dropped from any further cost-effectiveness comparisons because it was less cost-effective than the other comparator.

We created cost-effectiveness acceptability curves in order to assess the imprecision surrounding the estimates of costs, effects and cost-effectiveness. This approach involved drawing bootstrapped samples, with replacement, of the mean costs and mean QALYs from the original trial data. We repeated this process increasing the number of replications until the results were stable. This was achieved at 1000 replications. After using the new values generated from the bootstrapping exercise to calculate the difference in costs and effects between groups, we combined this information with a range of willingness to pay (WTP) values (£0, £10 000, £20 000, £30 000, £50 000) per QALY gained. This involved using the net benefit statistic[24] which multiplied the gain in health (QALYs) by the chosen WTP value, the incremental cost was then subtracted to obtain the net monetary benefit. We used

these results to generate a cost-effectiveness acceptability curve which graphically represented the probability of each of the interventions being cost-effective at each of the prespecified value for society's WTP for a QALY.[25] All analyses were carried out in Stata V.15.[26]

### Sensitivity, subgroup and per-protocol analyses

We conducted deterministic sensitivity analyses in order to assess the robustness of the cost-effectiveness results for three scenarios.

First, we examined the impact of assigning a value of zero to missing total healthcare costs.

Second, we examined the possibility that those participants with missing total healthcare costs may have used some services and hence incurred some costs. Under this scenario, once we established that information was missing at random, we applied truncated normal regression methods excluding total costs values below zero and above £25 000 (as the highest observed value for total costs) and used age, sex and baseline ARAT as covariates.

Finally, we investigated whether increasing the life span of the MIT-Manus robotic gym system from 5 to 7 years would affect the cost-effectiveness of the interventions.

In order to explore the impact of time since stroke on the cost-effectiveness of the interventions, we conducted an exploratory analysis for the subgroups outlined in the protocol.[4] Three subgroups were prespecified (<3 months, 3–12 months and >12 months) according to time since stroke.

A secondary per-protocol cost-effectiveness analysis removing from the data set those participants who did not receive at least 20 sessions of therapy in the robot-assisted training and the EULT programme groups was also conducted. The cut-off point of 20 sessions was based on clinical evidence that an additional 20 hours of therapy compared with control interventions leads to improvements on functional outcome.[27]

We combined each sensitivity, subgroup and per-protocol analyses with bootstrapping in order to reflect the imprecision surrounding the cost-effectiveness results.

### Extrapolation of trial results

We carried out a modelling exercise designed to extrapolate the mean QALYs at 6 months to 12 months based on the results of the trial, for this, we made a number of assumptions. First, we assumed that participants across all groups maintained the same utility levels reported at 6 months postrandomisation. Second, we considered all intervention costs and therapy-related costs (physiotherapy, occupational therapy, and speech and language therapy) as sunk costs, since they were deemed not to continue beyond the 6-month trial period. However, we assumed that all other levels of healthcare resource use remained constant from 6 to 12 months postrandomisation.

We calculated the difference in mean costs and effects between the randomised groups with differences across groups not being formally tested. We used seemingly unrelated regression[23] methods to calculate the adjusted ICER at 12 months including the same explanatory variables used in the base-case cost-effectiveness analysis. As in the base-case analysis we used the bootstrapping technique to present uncertainty surrounding mean costs, effects and cost-effectiveness.

### Patient and public involvement

We have designed and reported our research with input from stroke survivors. Members from the North East Stroke Research Network Patient and Carer Panel provided input to the design and content of trial documents, including health economics questionnaires.

## RESULTS

### Completeness of data

Most participants (96%) across all groups completed the health service utilisation questionnaire at baseline. There was a progressive increase in non-responses to the health service utilisation questionnaires over the 6-month follow-up period. The pattern of responses was similar across the intervention groups; however, the loss to follow-up was more pronounced in the usual care group. While completion rates were 83% for the robot-assisted training group and 85% for the EULT group, this decreased to 70% of usual care participants. Completion rate of EQ-5D-5L at baseline was 99% across all of participants. This decreased to 88% at 3 months and 82% at 6 months. The highest number of non-responders belonged to the usual care group with response rates of 81% at 3 months and 75% at 6 months.

### Health service use and costs

Reported health service resource use was broadly similar across all randomised groups at 6 months (table 1). Large SD indicate that there was substantial variation in use of service between individuals in all three randomised groups, with a few participants reporting very high use of some services. While not statistically tested, the main apparent difference between groups was in the therapy services received, with usual care participants receiving more home physiotherapy and speech and language therapy sessions compared with the robotic-assisted training and EULT groups. Participants in usual care had a higher reported mean number of contacts with general practice and nursing services compared with the robotic-assisted training and EULT groups.

The mean total cost per participant for each randomised group is reported in table 2. We also report the mean total cost per cost category per participant. The highest mean costs per participant were associated with the use of social care services which included stays in residential and nursing home facilities and care assistance received at home. The average cost per participant was higher in the usual care group in all categories except in secondary care and other NHS and social services used by the participants. The addition of the intervention costs; however, reversed this finding making the mean cost per

**Table 1** Reported health service resource use at 6 months

| Area of resource utilisation | Robot-assisted training (n=257) | | EULT (n=259) | | Usual care (n=254) | |
|---|---|---|---|---|---|---|
| | Respondents n* | Mean contacts (SD) | Respondents n* | Mean contacts (SD) | Respondents n* | Mean contacts (SD) |
| GP surgery | 205 | 1.80 (3.88) | 208 | 1.49 (2.08) | 168 | 1.83 (2.17) |
| GP home | 209 | 0.37 (1.08) | 213 | 0.27 (0.73) | 174 | 0.38 (1.29) |
| GP phone | 207 | 0.53 (1.22) | 207 | 0.39 (0.96) | 172 | 0.30 (0.77) |
| Nurse surgery | 203 | 0.61 (1.39) | 210 | 0.49 (1.19) | 169 | 0.90 (5.58) |
| Nurse home | 206 | 0.47 (2.07) | 213 | 0.45 (1.65) | 170 | 5.39 (44.63) |
| Nurse phone | 211 | 0.08 (0.56) | 212 | 0.52 (0.31) | 174 | 0.04 (0.25) |
| NHS direct | 211 | 0.11 (0.44) | 213 | 0.08 (0.35) | 174 | 0.03 (0.32) |
| Physiotherapy hospital | 210 | 2.18 (6.10) | 211 | 2.99 (9.23) | 171 | 2.64 (7.72) |
| Physiotherapy home | 205 | 2.76 (8.76) | 207 | 3.10 (8.04) | 170 | 4.35 (11.87) |
| Physiotherapy at general practice surgery | 213 | 0.13 (1.16) | 212 | 0.40 (2.63) | 176 | 0.50 (3.85) |
| Physiotherapy elsewhere | 212 | 0.23 (1.45) | 213 | 0.21 (1.82) | 175 | 0.62 (4.63) |
| Occupational therapy hospital | 211 | 0.64 (3.14) | 211 | 1.23 (7.23) | 174 | 0.57 (3.53) |
| Occupational therapy home | 210 | 1.56 (5.39) | 209 | 1.27 (4.28) | 173 | 1.95 (7.23) |
| Occupational therapy at general practice surgery | 213 | 0.00 (0.07) | 211 | 0.01 (0.14) | 176 | 0.02 (0.23) |
| Occupational therapy elsewhere | 212 | 0.16 (1.49) | 211 | 0.00 (0.00) | 176 | 0.00 (0.00) |
| Speech and language therapy hospital | 213 | 0.57 (2.79) | 213 | 0.65 (3.74) | 173 | 0.94 (5.24) |
| Speech and language therapy home | 210 | 0.47 (2.00) | 212 | 0.64 (3.97) | 175 | 2.22 (15.42) |
| Speech and language therapy at general practice surgery | 212 | 0.00 (0.00) | 213 | 0.03 (0.42) | 176 | 0.02 (0.13) |
| Speech and language therapy elsewhere | 213 | 0.05 (0.42) | 213 | 0.47 (0.68) | 175 | 0.03 (0.45) |
| A&E visits | 213 | 0.33 (0.77) | 214 | 0.37 (0.98) | 178 | 0.24 (0.70) |
| Outpatient appointments | 212 | 1.64 (3.14) | 215 | 1.42 (2.88) | 176 | 1.48 (4.24) |
| Hospital nights after being admitted via A&E | 213 | 0.79 (4.77) | 215 | 1.83 (12.95) | 176 | 0.70 (3.59) |
| Hospital nights NOT admitted via A&E | 213 | 0.28 (3.09) | 215 | 0.03 (0.19) | 176 | 0.25 (1.81) |
| Day patient treatment (half day) | 205 | 0.08 (0.35) | 210 | 0.09 (0.46) | 175 | 0.06 (0.32) |

Continued

**Table 1** Continued

| Area of resource utilisation | Robot-assisted training (n=257) | | EULT (n=259) | | Usual care (n=254) | |
|---|---|---|---|---|---|---|
| | Respondents n* | Mean contacts (SD) | Respondents n* | Mean contacts (SD) | Respondents n* | Mean contacts (SD) |
| Day patient treatment (full day) | 199 | 0.03 (0.17) | 201 | 0.02 (0.18) | 169 | 0.06 (0.07) |
| Residential care | 213 | 1.75 (14.65) | 215 | 2.08 (17.07) | 177 | 1.39 (10.72) |
| Nursing home | 213 | 0.00 (0.00) | 216 | 0.83 (12.24) | 177 | 3.08 (23.51) |
| Meals on wheels | 213 | 0.02 (0.27) | 213 | 0.04 (0.49) | 177 | 0.08 (1.05) |
| Home help personal care | 211 | 2.89 (6.77) | 210 | 3.04 (7.42) | 173 | 3.17 (7.03) |
| Home help household tasks | 213 | 0.74 (3.21) | 212 | 0.68 (3.87) | 175 | 1.06 (4.54) |
| Home help shopping | 213 | 0.11 (0.74) | 212 | 0.07 (0.53) | 176 | 0.11 (0.76) |
| Health visitor | 212 | 0.05 (0.45) | 213 | 0.09 (0.85) | 177 | 0.03 (0.28) |
| Geriatrician | 212 | 0.03 (0.42) | 213 | 0.02 (0.23) | 177 | 0.00 (0.00) |
| Psychiatrist | 212 | 0.13 (0.72) | 213 | 0.75 (0.54) | 175 | 0.01 (0.11) |
| Psychologist | 209 | 0.45 (2.07) | 211 | 0.28 (1.14) | 176 | 0.44 (2.37) |
| Chiropodist | 210 | 0.55 (1.19) | 210 | 0.39 (1.06) | 173 | 0.44 (1.02) |
| Optician | 210 | 0.25 (0.56) | 212 | 0.23 (0.53) | 174 | 0.32 (0.64) |
| Pharmacist | 211 | 0.66 (2.33) | 207 | 0.60 (2.28) | 174 | 1.16 (4.35) |

*n denotes the number of participants who completed all or part of the questionnaire.
A&E, accident and emergency; EULT, enhanced upper limb therapy; GP, general practitioner.

**Table 2** Total cost (£) over 6 months for all participants with full economic data

| Area of resource utilisation | Robot-assisted training (n=257) | | EULT (n=259) | | Usual care (n=254) | |
|---|---|---|---|---|---|---|
| | n | Mean (SD) | n | Mean (SD) | n | Mean (SD) |
| Intervention costs | 257 | 2872 (0) | 259 | 1399 (0) | 0 | – |
| Primary care costs and community-based healthcare (including therapy services) | 213 | 743 (1031) | 215 | 777 (1262) | 177 | 1078 (1813) |
| Social care | 213 | 1410 (3146) | 216 | 1541 (3943) | 178 | 1890 (4281) |
| Secondary care | 213 | 733 (2247) | 216 | 988 (4486) | 178 | 668 (1880) |
| Medication costs | 157 | 149 (302) | 162 | 154 (273) | 126 | 198 (347) |
| Other NHS and social services | 11 | 727 (983) | 13 | 790 (946) | 9 | 307 (406) |
| Deceased participants | 1 | 0 (0) | 3 | 13953 (4516) | 0 | – |
| Mean total cost | 257 | 5387 (4054) | 259 | 4451 (6033) | 178 | 3785 (5437) |
| Mean difference between robot-assisted training and usual care with 95% CI; p value | 1601 (706 to 2496); <0.001 | | | | | |
| Mean difference between EULT and usual care with 95% CI; p value | 665 (–444 to 1774); 0.239 | | | | | |

EULT, enhanced upper limb therapy; n, randomised n; NHS, National Health Service.

**Table 3** Utility scores at baseline, 3 months and 6 months and QALYs at 6 months

| Time period | Robot-assisted training (n*=257) | | EULT (n*=259) | | Usual care (n*=254) | |
| | n | Mean (SD) | n | Mean (SD) | n | Mean (SD) |
| --- | --- | --- | --- | --- | --- | --- |
| Baseline EQ-5D-5L score | 254 | 0.36 (0.26) | 259 | 0.39 (0.25) | 254 | 0.37 (0.26) |
| 3-month EQ-5D-5L score | 232 | 0.45 (0.27) | 236 | 0.48 (0.24) | 207 | 0.42 (0.29) |
| 6-month EQ-5D-5L score | 223 | 0.46 (0.29) | 222 | 0.50 (0.27) | 190 | 0.46 (0.27) |
| QALYs at 6 months after multiple imputation | 254 | 0.21 (0.12) | 259 | 0.23 (0.10) | 254 | 0.21 (0.11) |
| Mean difference in QALYs between robot-assisted training and usual care with 95% CI; p value | 0.00 (−0.20 to 0.20); 0.995 | | | | | |
| Mean difference in QALYs between EULT and usual care with 95% CI; p value | 0.02 (0.00 to 0.35); 0.080 | | | | | |

*n=number randomised.

EULT, enhanced upper limb therapy; QALY, quality adjusted life year.

participant highest in the group receiving robot-assisted training. We found some evidence the mean difference in costs between the robot-assisted training group and the usual care group was higher (mean difference: 1601 (95% CI 706 to 2496)). However, there was no evidence of a difference between the EULT group and usual care higher (mean difference: 665 (95% CI −444 to 1774)).

### Health outcomes
The mean utility scores across all randomised groups were similar at baseline, 3 months and 6 months as were mean QALYs (table 3). The EULT group had the highest mean QALY (0.23) followed by robot-assisted training and usual care groups, both of which had a mean QALY over 6 months of 0.21. The mean differences in QALYs between each of the intervention groups (robot-assisted training and EULT) and usual care were found to be very small and there was no evidence of a difference between randomised groups.

### Cost-effectiveness analysis, subgroup analysis, per-protocol analysis and longer-term model
The base-case cost-effectiveness analysis results (table 4) show the adjusted ICER was £74 100 for the comparison between EULT and usual care. Robot-assisted therapy was, on average, dominated by EULT since it was both, on average, more costly and less effective. The cost-effectiveness acceptability curve (figure 1), shows that EULT had a 19% chance of being cost-effective at the £20 000 WTP threshold value. The probability of EULT being cost-effective remained below 40% at all the WTP values considered in the analysis. Robot-assisted training had no probability of being cost-effective at all WTP values considered in the analysis.

The results from the subgroup analysis are also summarised in table 4. These showed that robot-assisted training remained dominated on average by EULT in both

the adjusted and unadjusted results for all subgroups. The highest ICER (£126 143) comparing EULT with usual care was linked to those participants who had a stroke more than 12 months before randomisation. The subgroup of participants who were less than 3 months poststroke at randomisation had the lowest ICER (£31 400) for the comparison of EULT with usual care. The bootstrapped sensitivity analysis for this group suggested that EULT had a 41% probability of being cost-effective at the £20 000 WTP threshold.

Results from the per-protocol analysis did not change the direction of the cost-effectiveness results. Usual care remained the least costly option followed by EULT and robot-assisted training. The ICER for EULT and usual care was £68 000 and EULT only had a 17% probability of being cost effective at the £20 000 WTP threshold. The probability of robot-assisted therapy being considered cost-effective was very low.

Table 4 shows the results from the economic model which extrapolated the trial data on costs and effects to 12 months. Unadjusted mean costs per participant were lowest in the EULT group (£6892) followed closely by usual care (£6916) and robot-assisted training (£7538). Mean QALYs were in line with those seen in the base-case analysis with participants in the EULT group having highest mean QALYs at 12 months (0.48) followed by usual care (0.47) and robot-assisted training (0.44). Once we adjusted for baseline costs, baseline utility score, study centre, randomised group and time since stroke, usual care reverted to being the least costly option. The ICER for the comparison between EULT and usual care was £6095, however there was only a 55% probability of EULT being considered cost-effective compared with usual care at the £20 000 WTP value. Robot-assisted training had no probability of being cost-effective at this WTP value.

**Table 4** Results from base-case cost-effectiveness analysis, subgroup analyses and longer-term economic model

| Scenario | Robot-assisted training (n=257) | EULT (n=259) | Usual care (n=254) | ICER | Probability of each therapy being cost-effective at the £20 000 WTP threshold | | |
| --- | --- | --- | --- | --- | --- | --- | --- |
| | | | | | Robot-assisted training | EULT | Usual care |
| Base-case analysis | | | | | | | |
| Cost—£, unadjusted, mean (CI)* | 5387 (4777 to 5996) | 4451 (3548 to 5354) | 5387 (4777 to 5996) | – | – | – | – |
| QALY— unadjusted, mean (CI)* | 0.21 (0.195 to 0.229) | 0.23 (0.213 to 0.244) | 0.21 (0.194 to 0.230) | – | – | – | – |
| ICER (£ per QALY)— adjusted EULT vs usual care† | – | – | – | 74 100 | 0.00 | 0.19 | 0.81 |
| Subgroup analysis (less than 3 months—time since stroke) | | | | | | | |
| Cost—£, unadjusted, mean (CI) | 5485 (3938 to 7032) | 3863 (2527 to 5199) | 3328 (1443 to 5213) | – | – | – | – |
| QALY— unadjusted, mean (CI)* | 0.22 (0.19 to 0.25) | 0.24 (0.20 to 0.28) | 0.21 (0.17 to 0.25) | – | – | – | – |
| ICER (£ per QALY)— adjusted EULT vs usual care† | – | – | – | 31 400 | 0.00 | 0.41 | 0.59 |
| Subgroup analysis (3 to 12 months—time since stroke) | | | | | | | |
| Cost—£, unadjusted, mean (CI)* | 5790 (4793 to 6786) | 5084 (3393 to 6774) | 4943 (3228 to 6658) | – | – | – | – |
| QALY— unadjusted, mean (CI) | 0.20 (0.18 to 0.23) | 0.23 (0.20 to 0.25) | 0.21 (0.19 to 0.24) | – | – | – | – |
| ICER (£ per QALY)— adjusted EULT vs usual care† | – | – | – | 79 400 | 0.04 | 0.37 | 0.59 |
| Subgroup analysis (more than 12 months) | | | | | | | |
| Cost—£, unadjusted, mean (CI)* | 4822 (4036 to 5728) | 3961 (2783 to 5138) | 2823 (1299 to 4348) | | – | – | – |
| QALY— unadjusted, mean (CI)* | 0.22 (0.18 to 0.25) | 0.23 (0.20 to 0.25) | 0.21 (0.18 to 0.24) | – | – | – | – |
| ICER (£ per QALY)— adjusted EULT vs usual care§ | – | – | – | 126 143 | 0.01 | 0.15 | 0.84 |
| Per-protocol analysis | | | | | | | |
| Cost—£, unadjusted, mean (CI)* | 5595 (4929 to 6261) | 4551 (3596 to 5501) | 3785 (2801 to 4770) | – | – | – | – |

Continued

**Table 4** Continued

| Scenario | Robot-assisted training (n=257) | EULT (n=259) | Usual care (n=254) | ICER | Probability of each therapy being cost-effective at the £20 000 WTP threshold | | |
| --- | --- | --- | --- | --- | --- | --- | --- |
| | | | | | Robot-assisted training | EULT | Usual care |
| QALY—unadjusted, mean (CI)* | 0.22 (0.20 to 0.24) | 0.23 (0.21 to 0.25) | 0.21 (0.19 to 0.23) | – | – | – | – |
| ICER (£ per QALY)—adjusted EULT vs usual care† | – | – | – | 68 000 | 0.00 | 0.17 | 0.83 |
| Extrapolation to 12 months | | | | | | | |
| Cost—£, unadjusted, mean (CI)* | 7538 (6350 to 8725) | 6892 (5149 to 8635) | 6916 (5003 to 8830) | – | – | – | – |
| QALY—unadjusted, mean (CI)* | 0.44 (0.40 to 0.48) | 0.48 (0.44 to 0.51) | 0.45 (0.41 to 0.48) | – | – | – | – |
| ICER (£ per QALY)—adjusted EULT vs usual care† | – | – | – | 6095 | 0.10 | 0.55 | 0.35 |

*98.33% CI used throughout the analyses for the three-arm comparison.
†Data adjusted for baseline costs, baseline utility score, study centre, randomised group and time since stroke.
EULT, enhanced upper limb therapy; ICER, incremental cost–effectiveness ratio; QALY, quality adjusted life year; WTP, willingness to pay.

## Sensitivity analyses

The different scenarios explored in these analyses did not change the direction of the results from the base-case cost-effectiveness results. Robot-assisted training remained dominated on average by EULT in all instances. First, when we changed the missing costs to zero, the resulting ICER between EULT and usual care increased to £172 000. This increase is to be expected since all participants with missing total costs belong to the usual care group. By imputing zero for missing costs the mean costs for the usual care participants decreased and hence, when this was done, the ICER increased when compared with EULT.

Second, applying multiple imputation methods to missing costs resulted in an increase to the unadjusted mean usual care costs (£4451) compared with the base-case results (£3785). Consequently, the resulting ICER from the comparison between EULT and usual care decreased to £50 000 with the probability of EULT being cost-effective at £20 000 increasing to 27%.

Third, extending the life of the robotic gym system resulted in a reduction of the mean capital costs per patient and hence, in a lower mean total cost for the robot-assisted training group (£5085) compared with the base-case analysis (£5387). There were no changes to the utility scores across all groups nor to the mean costs for EULT and usual care, hence the resulting ICER for the comparison of EULT and usual care remained the same as in the base-case analysis (£74 100).

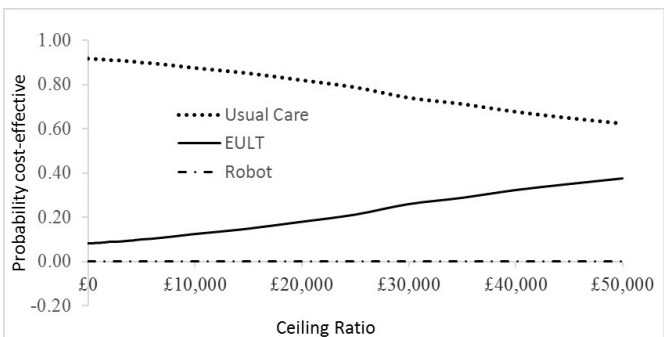

**Figure 1** Cost-effectiveness acceptability curve (base-case analysis)—adjusted bootstrapped replications for cost-effectiveness analysis. EULT, enhanced upper limb therapy.

## DISCUSSION

The RATULS trial found no evidence that robot-assisted training, as delivered in the study, improved upper limb function 'success' for patients who had a stroke with moderate or severe upper limb functional limitation when compared with a EULT programme or usual care.[5] Our economic evaluation strengthens the evidence base of these upper limb rehabilitation programmes through the evaluation of their cost-effectiveness.

Results from the base-case cost-effectiveness analysis suggested that, on average, robot-assisted training was more costly than both EULT and usual care and that robot-assisted training was slightly less effective than EULT. EULT was on average more costly and as effective as usual care in the unadjusted analyses and more costly and more effective in the adjusted analyses. The balance of probabilities favoured usual care as the preferred upper limb rehabilitation therapy over the range of WTP values considered. Focusing on the society's WTP for a QALY, the bootstrapped analysis suggested that, EULT, as delivered in this trial, was unlikely to be cost-effective over any of the WTP values considered, despite being more effective than robot-assisted training and usual care. The subgroup, sensitivity and per-protocol analyses did not change the direction of the base-case cost-effectiveness results. Extrapolating within-trial results to 12 months produced the lowest ICER for the comparison between EULT and usual care overall, however, high uncertainty surrounds the assumptions made about how costs and utilities change beyond the trial follow-up.

The main strength of this economic evaluation is that it was conducted as part of a rigorously run RCT and followed guidelines for best practice throughout.[11 28] As a result, we were able to base the economic evaluation on individual patient data collected during the trial and benefited from low levels of missing healthcare resource use and quality of life data. However, the loss to follow-up in the usual care group may have led to an underestimation of resource use for these participants. Nevertheless, when we imputed missing values the results were still consistent with those from the base-case analysis.

One of the main challenges in conducting the economic evaluation was the difficulty to ascertain the specific components of usual care therapy. Log books designed to capture detailed usual care information were completed by participants. The information gathered was to be used alongside the health service utilisation questionnaire. Completion rates, however, were very low and we were unable to incorporate these data into the economic evaluation. We overcame this by drawing on the information captured via our primary data collection tool, the health-care service utilisation questionnaire, where participants recorded any therapy sessions received during the trial period.

Through the collection of self-reported quality of life information at three points during the study using the EQ-5D-5L questionnaire,[19] we were able to measure quality of life gains for participants across all groups. One strength of this generic tool is that decision makers will be able to make priority-setting decisions not only for this patient group but across different disease areas.[11] However, the EQ-5D-5L questionnaire does not capture transitory changes as it only asks about health on the day the participant completes it. In addition, this questionnaire and the QALYs derived from it are not stroke specific and it is unknown whether we were able to accurately capture changes in quality of life in this patient group.

A noteworthy limitation of the economic evaluation is associated with the timeframe of the trial. The within-trial economic evaluation assessed the cost-effectiveness of the interventions at 6 months. A longer-term perspective was originally planned but due to limitations of the data, extrapolation to 12 months only was conducted. The results however, need to be interpreted with caution due to the assumptions made on both costs and utility values.

The economic evaluation fills a significant evidence gap with this being the first economic evaluation comparing robot-assisted training with usual care having been conducted in the UK NHS setting. This evaluation expands the analysis conducted on the cost-effectiveness of the MIT-Manus robotic system as part of the VA Robotics trial.[6 7] This study, confined to the US healthcare system, reported a small QALY gain for the robot group compared with usual care and not significant differences in costs between groups. Differences in the healthcare system between both countries means that our economic evaluation is key for making cost-effectiveness results relevant to the UK NHS setting. Furthermore, it takes into account a number of key differences in the design of both studies. First, while the VA robotics trial assumed that the robotic gym could be used simultaneously by two patients, the RATULS trial was designed to deliver robot-assisted training on a one-to-one basis. Second, the components of the intensive comparison therapy differed in each study. Third, we used E5-5D-5L as the recommended tool to calculate QALYs, while the VA robotic trial calculated QALYs from responses to the Health Utility Index Mark 3 questionnaire.[29] All these points reduce the grounds for comparability between studies and supports the need for the economic analysis we conducted alongside the RATULS trial.

The use of multiple sites contributed to the generalisability of the economic evaluation. The analyses controlled for differences in sites hence minimising the chance of obtaining biased results from differences in costs and effects driven by location.

In conclusion, robot-assisted training was not found to be cost-effective in comparison to EULT and usual care. This economic evaluation suggested that usual care remained the most cost-effective type of upper limb rehabilitation compared with a EULT therapy programme and robot-assisted training for patients who had a stroke with moderate or severe functional limitation.

The results create opportunities for further research. In particular, further research could explore the potential effect on both costs and QALYs from reconfigurations to the delivery of EULT and robot-assisted training. It remains unclear, for example, whether delivering therapy in a group setting may first, be feasible in the NHS setting and second improve the quality of life and clinical outcomes. Further development of

these interventions may increase their cost-effectiveness compared with usual care. Additionally, studies with a longer follow-up data may help establish whether the QALY gains derived from the interventions are sustained beyond the set timeframe of the trial, this can then lead to a robust assessment of their long-term cost-effectiveness.

**Author affiliations**
[1]Population Health Sciences Institute, Newcastle University Faculty of Medical Sciences, Newcastle upon Tyne, UK
[2]Stroke Research Group, Population Health Sciences Institute, Newcastle University Faculty of Medical Sciences, Newcastle upon Tyne, UK
[3]Stroke Northumbria, Northumbria Healthcare NHS Foundation Trust, North Shields, UK
[4]Stroke Medicine, Barking Havering and Redbridge Hospitals NHS Trust, Romford, UK
[5]Northwick Park, London North West University Healthcare NHS Trust, Harrow, UK
[6]Institute of Cardiovascular and Medical Sciences, University of Glasgow, Glasgow, UK
[7]Nursing, Midwifery and Health, Northumbria University, Newcastle upon Tyne, UK
[8]Oxford Academic Health Science Network, Oxford University Hospitals NHS Foundation Trust, Oxford, UK
[9](Lay Investigator) Contact Stroke Research Group, Population Health Sciences Institute, Newcastle University Faculty of Medical Sciences, Newcastle upon Tyne, UK
[10]Queen Elizabeth University Hospital, NHS Greater Glasgow and Clyde, Glasgow, UK
[11]Department of Mechanical Engineering, Massachusetts Institute of Technology, Cambridge, Massachusetts, USA
[12]School of Health Sport and Bioscience, University of East London, London, UK
[13]School of Health and Life Sciences, Glasgow Caledonian University, Glasgow, UK
[14]School of Pharmacy, University of Sunderland, Sunderland, UK

**Acknowledgements** We would like to thank Jenni Hislop for her input into the health economics analysis plan and the development of the data collection tools.

**Contributors** CF-G conducted the health economic evaluation. LV and LT are senior health economists and were involved in the main study design, delivery, analysis and interpretation of the data and drafting the manuscript. TMH contributed to the health economic evaluation and drafting of the manuscript. HR was the chief investigator of the study. HB, HIK, FVW, GF, LS, LA, TF, CP, DT, SW, JD were involved in the main study design, delivery, data management, data analysis and drafting the manuscript. NW, SA, DC, RF, NH contributed to study delivery, interpretation of the data and drafting the manuscript. SH was a coinvestigator and is a service user, he was involved in the main study design and delivery, interpretation of data. All authors have commented upon drafts of the manuscript and have given final approval to this version.

**Funding** This work was supported by the National Institute for Health Research Health Technology Assessment Programme (reference: 11/26/05). The views and opinions expressed here are those of the authors and do not necessarily reflect those of the HTA programme, NIHR, the UK National Health Service (NHS) or UK Department of Health.

**Competing interests** HR reports grants from NIHR, during the conduct of the study; personal fees from Bayer, outside the submitted work; and Member of NIHR HTA CET panel 2010–2014. GF reports grants from National Institute for Health Research, during the conduct of the study; personal fees from Amgen, personal fees from Daiichi Sankyo, grants and personal fees from Medtronic, personal fees from Stryker, personal fees from Pfizer, personal fees from Bayer, outside the submitted work. CP and LS report grants from National Institute for Health Research, during the conduct of the study. HIK reports other from Interactive Motion Technologies, other from 4Motion Robotics, outside the submitted work; in addition, HIK has a patent Interactive Robotic Therapist; US Patent 5466213; 1995; Massachusetts Institute of Technology issued, and a patent Wrist And Upper Extremity Motion; US Patent No. 7618381; 2009; Massachusetts Institute of Technology licensed to Bionik Laboratories.

**Patient and public involvement** Patients and/or the public were involved in the design, or conduct, or reporting, or dissemination plans of this research. Refer to the Methods section for further details.

**Patient consent for publication** Not required.

**Ethics approval** Ethical approval was obtained for the clinical trial and the economic questionnaires. Approval was granted by the National Research Ethics Committee Sunderland (reference 12/NE/0274). The research here submitted is confined to the within trial economic evaluation.

**Provenance and peer review** Not commissioned; externally peer reviewed.

**Data availability statement** Data are available upon reasonable request. De-identified participant data will be made available to scientific researchers upon approval of their study protocol and analysis plan, by a committee of the RATULS team. Proposals should be directed to the corresponding author. A data sharing agreement will need to be signed by data requestors.

**ORCID iDs**
Cristina Fernandez-Garcia http://orcid.org/0000-0002-7113-225X
Nina Wilson http://orcid.org/0000-0001-5908-1720

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
