## [Reviewer comments · BMJ Open]

ARTICLE DETAILS

TITLE (PROVISIONAL)	Economic evaluation of robot-assisted training versus an enhanced upper limb therapy programme or usual care for patients with moderate or severe upper limb functional limitation due to stroke: results from the RATULS randomised controlled trial
AUTHORS	Fernandez-Garcia, Cristina; Ternent, Laura; Homer, Tara; Rodgers, Helen; Bosomworth, Helen; Shaw, Lisa; Aird, Lydia; Andole, Sreeman; Cohen, David; Dawson, Jesse; Finch, Tracy; Ford, Gary; Francis, Richard; Hogg, Steven; Hughes, Niall; Krebs, H; Price, Christopher; Turner, Duncan; Van Wijck, Frederike; Wilkes, Scott; Wilson, Nina; Vale, Luke

VERSION 1 – REVIEW

REVIEWER	Adelaida María Castro Sánchez University of Almeria, Spain
REVIEW RETURNED	02-Aug-2020

GENERAL COMMENTS	The manuscript titled “Economic evaluation of robot-assisted training versus an enhanced upper limb therapy programme or usual care for patients with moderate or severe upper limb functional limitation due to stroke: results from the RATULS randomised controlled trial” can be accepted for publish in BMJ Open following next comments: - Abstract acronyms must be specified.- Specify centers in Setting.- Specify statistical significance data in abstract.- The first two strengths can be merged, and a cost-effectiveness strength could be added.- Detail the calculation of the sample size, and the selection of the sample for each group.- Detail more specifically how the effect of the intervention was determined.- Detail more explicitly the calculations in relation to the incremental cost-utility, and the profitability coefficients.
--

REVIEWER	Louise Craig University of Glasgow
REVIEW RETURNED	23-Feb-2021

GENERAL COMMENTS	Thank you for the opportunity to review this paper. One of the key strengths of this EE is being the conduct alongside a clinical trial to ensure an individual patient data approach to costing data. The introduction and the discussion require some re-writing and there are a few areas that the reporting needs to be clearer. Further points are raised below -
---

	Introduction  1. The importance for EE generally should be considered i.e. rehab. is highly resource-intensive, allows policy makers to consider the potential trade-offs between all relevant costs and benefits 2. I found this slightly repetitive with the discussion especially the final paragraph beginning Line 70 3. The sentence 'After conducting a scoping review, we found little evidence of cost-effectiveness studies in the UK' needs to be completed. EE in UL may be limited so it may be that the authors have to widen this scoping to stroke rehabilitation generally such as intensive vs TAU. Limitations in previous research could be used to set the scene and allow the readers to fully appreciate the value of this EE. One possible paper that could be helpful is 'Approaches to economic evaluations of stroke rehabilitation', Craig et al 4. Editing required – 'the cost-effectiveness the MIT-Manus'..... Methods  1. Please state and justify the type of economic evaluation used 2. Please explicitly state the primary outcome measure(s) for the economic evaluation 3. Provide justification for perspective - as the potential outcomes of stroke rehabilitation are likely to be wide-reaching with implications for patients, carers and society it would be good to understand why NHS perspective was adopted. 4. PPI – please state if and what this input was specific to the EE. Results  1. Reword 'with usual care being the heaviest'..... 2. Table 1 – should the header row read 'Mean contacts' for all three groups? 3. Use the same headings and order of presentation as in the methods section Discussion  1. Whilst appreciating the use of QALYs allows for comparison it would be worth outlining the limitations of the QALY i.e. not capturing all the broader benefits that are associated with interventions with more levels of complexity and the short 6 mth f/up timeframe 2. Please reword this sentence as it is currently unclear what type of therapy the authors are referring to "It remains unclear, for example whether delivering therapy in a group or classroom....." Line 376
--	--

VERSION 1 – AUTHOR RESPONSE

Response to reviewers' comments for BMJ Open Submission

Reviewer 1			
5	Abstract acronyms must be specified	Abstract has now been amended and all acronyms have been specified.	Page 2

6	Specify centers in Setting	Abstract now includes details of all study centres.	Page 2/ Lines 30-31
7	Specify statistical significance data in abstract.	Results section in the abstract has now been amended as follows (changes in bold) to include information about statistical significance. P values have also been added to the results tables 2 &3 for ease: At six months, on average usual care was the least costly option (£3,785) followed by EULT (£4,451) with robot-assisted training being the most costly (£5,387). The mean difference in total costs between the usual care and robot-assisted training groups (£1,601) was statistically significant (p < 0.001). Mean QALYs were highest for the EULT group (0.22) but no evidence of a difference (p = 0.995) was observed between the robot-assisted training (0.21) and usual care groups (0.21). The incremental cost per QALY at six months for participants randomised to EULT compared to usual care was £74,100. Cost-effectiveness acceptability curves showed that robot-assisted training was unlikely to be cost-effective and that EULT had a 19% chance of being cost-effective at the £20,000 willingness to pay (WTP) threshold. Usual care was most likely to be cost-effective at all the WTP values considered in the analysis	Page 2/ Lines 45-54 Page 10/ Table 2 Page 10&11/Table 3
8	The first two strengths can be merged, and a cost-effectiveness strength could be added	Please see response to comment 2	Page 3/ Strengths and Limitations box

9	Detail the calculation of the sample size, and the selection of the sample for each group	The following text has been added to trial overview section: “The sample size calculation yielded a target sample size of 762 participants with 216 participants in each group required to provide 80% power at a significance level of 1.7%. The sample size was revised after protocol publication to 770 to allow for 15% attrition (rather than 10% as originally specified in the published protocol). Full details on sample size calculation and trial methodology have been reported elsewhere.”	Page 4/ Lines 90-94
10	Detail more specifically how the effect of the intervention was determined	The following text has been added to the trial overview section: “The definition of success differed depending on baseline severity of upper limb functional limitation. Success for a baseline ARAT score 0–7 was defined as an improvement of 3 points or more; a baseline ARAT 8–13 required an improvement of 4 points or more; baseline ARAT 14–19 required an improvement of 5 points or more; and finally a baseline ARAT 20–39 required an improvement of 6 points or more”	Page 4 / Lines 106-111
11	Detail more explicitly the calculations in relation to the incremental cost-utility, and the profitability coefficients	We apologise if this were not clear. The methods we have used are commonly employed but we realise that this is not necessarily clear for the general reader. Therefore, we have revised and expanded the text. In particular we have clarified the calculation of the ICER and expanded on how the cost-effectiveness acceptability curves were calculated. “We calculated mean costs and effects along with corresponding standard deviations. Where we report differences in mean costs and effects between all three randomised groups we used 98.33% confidence intervals, as this was a three arm comparison. We conducted all pairwise comparisons using 95% confidence intervals. Using seemingly unrelated regression modelling methodology in the adjusted cost-effectiveness analysis, we	Page 6 / Lines 174-201

		derived the incremental cost per gained QALY for each participant at six months. This approach involved estimating two linear regressions with their own dependent variable for costs and QALYs and a set of explanatory variables. We used randomised group, study centre, and time since stroke as explanatory variables for both costs and QALYs. Additionally, we incorporated baseline utility scores as an explanatory variable for the QALY equation and total baseline costs as an explanatory variable to the costs equation. We presented the results from the adjusted analysis in the form of an incremental cost-effectiveness ratio (ICER). The ICER is the difference in mean costs divided by the difference in mean effects (in this case QALYs) between two alternatives. In the analysis if a comparator was both more costly and less effective than the others it was dropped from any further cost-effectiveness comparisons because it was less cost-effective than the other comparator. We created cost-effectiveness acceptability curves in order to assess the imprecision surrounding the estimates of costs, effects and cost-effectiveness. This approach involved drawing bootstrapped samples, with replacement, of the mean costs and mean QALYs from the original trial data. We repeated this process increasing the number of replications until the results were stable. This was achieved at 1,000 replications. After using the new values generated from the bootstrapping exercise to calculate the difference in costs and effects between groups, we combined this information with a range of willingness to pay (WTP) values (£0, £10,000, £20,000, £30,000, £50,000) per QALY gained. This involved using the net benefit statistic which multiplied the gain in health (QALYs) by the chosen WTP value, the incremental cost was then subtracted to obtain the net monetary benefit. We used these results to generate a cost-effectiveness acceptability curve which graphically represented the probability of each of the interventions being cost-effective at each of the pre-specified value for society's WTP for a QALY. All analyses were carried out in Stata 15."	
--	--	--	--

Reviewer 2			
	Introduction		
12	The introduction and the discussion require some re-writing and there are a few areas that the reporting needs to be clearer. Further points are raised below -	First paragraph has been revised to improve readability as follows, changes in bold: “Stroke is the fourth leading cause of death in the UK and a leading cause of disability. Almost two thirds of stroke patients leaving leave hospital with a disability. A common disability following a stroke is loss of upper limb function. This results in a reduction of the individual’s autonomy and impedes activities of daily living. Approximately 80% of people with acute stroke have upper limb motor impairment, with 50% of patients still experiencing problems after four years following the stroke.”	Page 3/ Lines 63-67
13	The importance for EE generally should be considered i.e. rehab. is highly resource-intensive, allows policy makers to consider the potential trade-offs between all relevant costs and benefits	Many thanks for your comment and details of very useful paper which has now been cited in our manuscript. The revised final paragraph of introduction with new additions in bold is described below: After conducting a scoping review, we found little evidence of cost-effectiveness studies in the UK. The only economic evaluation we found in the literature assessed the cost-effectiveness of robot-assisted training therapy for upper limb rehabilitation within the US based VA Robotics study. This randomised controlled trial also assessed the cost-effectiveness of the MIT-Manus robotic gym system in upper limb rehabilitation in stroke patients. Given the resource intensive nature of stroke rehabilitation programmes⁸ and the lifelong impacts of stroke, evidence on the cost-effectiveness of these programmes derived from well designed economic evaluations is needed. The RATULS trial is, to our knowledge, the largest and first multicentre trial with sufficient statistical power to compare robot-assisted training with another evidence based therapy programme, or usual care.⁹ This paper reports the	Page 4 / Lines 76-85

		results from a within-trial analysis that formed part of the RATULS trial.	
14	I found this slightly repetitive with the discussion especially the final paragraph beginning Line 70	The following text from the introduction has now been deleted: “However, differences in the healthcare system, in the configuration of the robotic gym and in the components of the comparison therapy meant that the results from this study lacked generalisability to the UK setting.”	Removed from Introduction
15	The sentence ‘After conducting a scoping review, we found little evidence of cost-effectiveness studies in the UK’ needs to be completed. EE in UL may be limited so it may be that the authors have to widen this scoping to stroke rehabilitation generally such as intensive vs TAU. Limitations in previous research could be used to set the scene and allow the readers to fully appreciate the value of this EE. One possible paper that could be helpful is ‘Approaches to economic evaluations of stroke rehabilitation’, Craig et al	Please see response to comment 13	Page 4 / Lines 76-85
16	Editing required – ‘the cost-effectiveness	This has now been changed to:	Page 4/Line 79

	the MIT-Manus'.....	"the cost-effectiveness of the MIT-Manus robotic gym"	
	Methods		
17	Please state and justify the type of economic evaluation used	Text added to Economic Evaluation Methods section: "We conducted an economic evaluation consisting of a cost-utility analysis using the QALY as the primary outcome measure following guidance for best practice in health technology appraisal."	Page 4/Lines 114-115
18	Please explicitly state the primary outcome measure(s) for the economic evaluation	See response to comment 17	Page 4/Lines 114-115
19	Provide justification for perspective - as the potential outcomes of stroke rehabilitation are likely to be wide-reaching with implications for patients, carers and society it would be good to understand why NHS perspective was adopted	Response to reviewer: Thank you for this very important comment. We agree that taking a wider perspective is important. We had planned to take a wider perspective as part of the economic evaluation. In order to do this we collected information on participant and carer related costs via a time and travel questionnaire. This questionnaire was completed by the participants and their carers, who provided details about their out-of-pocket expenses relating to their most recent GP and secondary care appointments. This included travel time, time spent at appointments, mileage, parking and other transport-related costs. Details relating to the main activities that participants and carers would otherwise be doing were also provided. However, completion of these questionnaires was low and although time and travel costs were analysed they were not included in the economic evaluation. The analysis of time and travel data was included as an Appendix in the HTA report: https://www.journalslibrary.nihr.ac.uk/hta/hta24540#/full-report	
	PPI – please state if and what this input was specific to the EE.	See response to comment 3	Page 7 / Lines 238-240
	Results		

	Reword 'with usual care being the heaviest'.....	Text now reads as follows (changes in bold): "Whilst not statistically tested, the main apparent difference between groups was in the therapy services received, with usual care participants receiving more home physiotherapy and speech and language therapy sessions compared with the robotic-assisted training and EULT groups."	Page 8 / Lines 257-261
	Table 1 – should the header row read 'Mean contacts' for all three groups?	Header row has now been modified to read "Mean contacts" for all groups.	Page 8 / Table 1
	Use the same headings and order of presentation as in the methods section	Response to reviewer: Thank you for your comment. We feel that a discussion of missing data should start the results section so that the reader is informed of the level of data used for the analysis from the beginning. All other results are in the same order as listed in the methods section.	
	Discussion		
	Whilst appreciating the use of QALYs allows for comparison it would be worth outlining the limitations of the QALY i.e. not capturing all the broader benefits that are associated with interventions with more levels of complexity and the short 6 mth f/up timeframe	Response to reviewer: We have acknowledged the limitation of using QALYs and amended the text as follows (changes in bold): "Through the collection of self-reported quality of life information at three points during the study using the EQ-5D-5L questionnaire, we were able to measure quality of life gains for participants across all groups. One strength of this generic tool is that decision makers will be able to make priority-setting decisions not only for this patient group but across different disease areas. However, the EQ-5D-5L questionnaire does not capture transitory changes as it only asks about health on the day the participant completes it. In addition, this questionnaire and the QALYs derived from it are not stroke specific and it is unknown whether we were able to accurately capture changes in quality of life in this patient group. "	Page 14/Lines 372-379

		A limitation linked to the short timeframe of the trial has been acknowledged in the discussion section: “A noteworthy limitation of the economic evaluation is associated with the timeframe of the trial. The within-trial economic evaluation assessed the cost-effectiveness of the interventions at six months. A longer term perspective was originally planned but due to limitations of the data, extrapolation to 12 months only was conducted. The results however, need to be interpreted with caution due to the assumptions made on both costs and utility value”	Page 14/380-384
	Please reword this sentence as it is currently unclear what type of therapy the authors are referring to “It remains unclear, for example whether delivering therapy in a group or classroom.....” Line 376	Removed the term classroom from text	Removed from Page 15/Line 410

VERSION 2 – REVIEW

REVIEWER	Castro-Sanchez, Adelaida M. Universidad de Almeria, Enfermería, Fisioterapia y Medicina
REVIEW RETURNED	04-Apr-2021
GENERAL COMMENTS	This manuscript can be accepted for publication
REVIEWER	Craig, Louise University of Glasgow

REVIEW RETURNED	07-Apr-2021
-------------

GENERAL COMMENTS	Thank you for your responses and edits. I believe the paper now provides more interesting discussion around the conduct of economic evaluation in rehabilitation.
---